# Development and Characterization of Hydroxyethyl Cellulose-Based Gels Containing Lactobacilli Strains: Evaluation of Antimicrobial Effects in In Vitro and Ex Vivo Models

**DOI:** 10.3390/ph16030468

**Published:** 2023-03-22

**Authors:** Marcela Almeida dos Santos de Sousa, Alexia Figueiredo Ferreira, Camila Caetano da Silva, Marcos Andrade Silva, Tamyris Alicely Xavier Nogueira Bazan, Cristina de Andrade Monteiro, Andrea de Souza Monteiro, Joicy Cortez de Sá Sousa, Luís Cláudio Nascimento da Silva, Adrielle Zagmignan

**Affiliations:** 1Laboratory of Microbial Pathogenesis Patogenicidade Microbiana, CEUMA University, São Luís 65075-120, Brazil; 2Laboratory of Odontology, CEUMA University, São Luís 65075-120, Brazil; 3Laboratory of Research and Study in Microbiology, Federal Institute of Education, Science and Technology of Maranhão (IFMA), São Luís 65030-005, Brazil; 4Laboratory of Applied Microbiology, CEUMA University, São Luís 65075-120, Brazil

**Keywords:** probiotics, wounds, infection, ex vivo model

## Abstract

This study aimed to develop a hydroxyethyl cellulose-based topical formulation containing probiotics and to evaluate its antimicrobial action using in vivo and ex vivo models. Initially, the antagonistic effects of *Lacticaseibacillus rhamnosus* ATCC 10863, *Limosilactobacillus fermentum* ATCC 23271, *Lactiplantibacillus plantarum* ATCC 8014 and *Lactiplantibacillus plantarum* LP-G18-A11 were analyzed against *Enterococcus faecalis* ATCC 29212, *Klebsiella pneumoniae* ATCC 700603, *Staphylococcus aureus* ATCC 27853 and *Pseudomonas aeruginosa* ATCC 2785. The best action was seen for *L. plantarum* LP-G18-A11, which presented high inhibition against *S. aureus* and *P. aeruginosa*. Then, lactobacilli strains were incorporated into hydroxyethyl cellulose-based gels (natrosol); however, only the LP-G18-A11-incorporated gels (5% and 3%) showed antimicrobial effects. The LP-G18-A11 gel (5%) maintained its antimicrobial effects and viability up to 14 and 90 days at 25 °C and 4 °C, respectively. In the ex vivo assay using porcine skin, the LP-G18-A11 gel (5%) significantly reduced the skin loads of *S. aureus* and *P. aeruginosa* after 24 h, while only *P. aeruginosa* was reduced after 72 h. Moreover, the LP-G18-A11 gel (5%) showed stability in the preliminary and accelerated assays. Taken together, the results show the antimicrobial potential of *L. plantarum* LP-G18-A11, which may be applied in the development of new dressings for the treatment of infected wounds.

## 1. Introduction

The skin is the largest organ of the human body; this fact makes it a tissue susceptible to trauma and injury, with a significant impact on the individual and on the health system in the treatment and rehabilitation process [1]. The main skin function is to protect the internal organs, preventing the entry of microorganisms and harmful agents that can be harmful to health, as well as protecting against water loss and ultraviolet radiation [2,3]. The continuous loss of skin can be caused by different situations, such as physical, chemical, mechanical, vascular, infectious, allergic, thermal trauma or even by surgical cutting [4]. 

As it is an organ highly exposed to various external situations, many cases of skin and soft tissue infections are caused by so-called ESKAPE pathogens (*Enterococcus faecium*, *Staphylococcus aureus*, *Klebsiella pneumoniae*, *Acinetobacter baumannii*, *Pseudomonas aeruginosa* and *Enterobacter* species) that belong to the bacteria resistant against almost all commonly used antibiotics [5]. Among them, *S. aureus* and *P. aeruginosa* are commonly isolated from chronic wounds and cause the increase in resistance to topical antibiotics causing serious damage to health agencies [6].

The indiscriminate use of antibiotics has caused high rates of bacterial resistance, which leads to a risk to the quality of human health, intensifying clinical conditions that are difficult to treat and the risk of hospital infections, both of which are considered public health problems [7]. Moreover, the side effects of conventional therapies and the costs of infected-wound treatment increase the relevance of the development of novel topical agents [8]. Thus, lactobacilli strains have been widely used for wound healing, scar reduction, as well as infections being treating in vitro and in vivo treatments [9,10,11]. In this context, the topical application of lactobacilli has been suggested as a new alternative for wound treatment, due to their immunomodulatory and healing abilities, in addition to exhibiting antagonistic effects against pathogens through competitive exclusion [12,13].

To convey these probiotics, there are several vehicles or pharmaceutical formulations, such as cream, gels, cream-gels, ointments. The vehicle choice is made according to the specifications to guarantee the bacterial viability (as a probiotic product) and product stability, with it being necessary to observe factors such as the chemical and physical stability of the preparation, the proper preservation against microbial contamination and the uniformity of the active ingredient used [14].

Polysaccharides-based gels, such as those from cellulose and its derivatives, have been used for the development of healing formulations due their high biocompatibility and adsorption capacity, promoting a humid environment for the wound [10,15,16]. An example is natrosol, a hydroxyethyl cellulose-based gel with non-ionic characteristics, biodegradability and biocompatibility. This low-cost vehicle can tolerate a wide pH variation, in addition to having already shown healing activity [17,18,19]. Thus, the incorporation of lactobacilli strains with antimicrobial activity in natrosol gel is an interesting alternative for the treatment of infected wounds.

Therefore, this study aimed to develop topical formulations with probiotics for application in wound treatment. For this, the antimicrobial actions of four lactobacilli strains (*Lacticaseibacillus rhamnosus* ATCC 10863, *Limosilactobacillus fermentum* ATCC 23271, *Lactiplantibacillus plantarum* ATCC 8014 and *Lactiplantibacillus plantarum* LP-G18-A11) were screened against *E. faecalis* ATCC 29212, *K. pneumoniae* ATCC 700603, *S. aureus* ATCC 27853 and *P. aeruginosa* ATCC 2785. The strain with a higher inhibitory potential was selected for the formulation of a gel, which was characterized, and its antimicrobial effects were analyzed using an ex vivo model of an infected wound with porcine skin.

## 2. Results

### 2.1. Antagonism Potential of Strains

Initially, the antimicrobial action of the four selected lactobacilli were evaluated against *E. faecalis* ATCC 29212, *K. pneumoniae* ATCC 700603, *S. aureus* ATCC 27853 and *P. aeruginosa* ATCC 2785, as shown in Table 1. All of the lactobacilli strains showed high inhibition capacity against *S. aureus* [inhibition zone (IZ) > 6 mm], where *L. plantarum* LP-G18-A11 showed higher action (IZ = 12.5 ± 0.2). Regarding *P. aeruginosa*, only *L. plantarum* LP-G18-A11 showed high inhibition capacity (IZ = 8.5 ± 0.1), while none of the tested strains inhibited the growth of *K. pneumoniae* and *E. faecalis*.

### 2.2. Antimicrobial Activity of Natrosol Gel with Lactiplantibacillus plantarum LP-G18-A11

Subsequently, two formulations of natrosol gel (1.5% *w*/*v*) were manipulated using *L. plantarum* LP-G18-A11 at 5% or 3%. The other strains were incorporated into the respective gels only at 5% (due the lower antimicrobial action when compared with LP-G18-A11). The antimicrobial actions of these gels were evaluated against *P. aeruginosa* and *S. aureus*. Both LP-G18-A11-gels exhibited inhibitory effects (Table 2), with higher action observed for the formulation at 5%. The gels formulated with the other lactobacilli strains did not inhibit the growth of *P. aeruginosa* and *S. aureus.*

### 2.3. Viability of Lactiplantibacillus plantarum LP-G18-A11 in Natrosol Gel

Based on the above results, the LP-G18-A11-incorporated gels were selected for the following assays. The gels were stored at 25 °C and 4 °C and their bacterial viability was analyzed up to 90 days. In the first 14 days, the bacteria remained viable in both formulations and storage conditions (variation of 7.6 to 8.9 Log CFU/mL). After this period, there was a progressive loss of viable cells, where the bacterial incorporated into the gels kept at 25 °C lost their viability after 21 days of storage. The gels that remained refrigerated showed bacterial counts between 5 and 6 Log CFU/mL up to 30 days. In addition, the 5% gel showed bacterial populations of 5 Log CFU/mL after 90 days of storage (Figure 1).

### 2.4. Time-Kill Curve

Given the best results obtained with the *L. plantarum* LP-G18-A11 at 5%, a time-kill analysis was performed using *S. aureus* as the model. Significant reductions (*p* < 0.05) were observed after 2 h of contact with the *L. plantarum* LP-G18-A11-incorporated gel. After 4 h, no growth was detected for *S. aureus* and this effect remained until 30 h (Figure 2). Furthermore, the *L. plantarum* LP-G18-A11 remained viable throughout the trial (~10 Log CFU/mL).

### 2.5. Antimicrobial Activity of LP-G18-A11-Incorporated Gel (5%) in Wound Ex Vivo Model

The antimicrobial effects of the LP-G18-A11-incorporated gel (5%) were further analyzed in an ex vivo model using porcine skin. In this assay, experimental wounds were performed in the porcine skin and contaminated with *S. aureus* and *P. aeruginosa*. After 24 h, the contaminated wounds were treated with the LP-G18-A11-incorporated gel (5%) and exhibited a reduced number of bacteria (*p* < 0.05). The reductions in the CFU counts were 31% and 52% for *S. aureus* and *P. aeruginosa*, respectively. These results were similar to those obtained with ciprofloxacin (37% and 62%, respectively) (Figure 3A,B).

After 72 h of treatment, no reduction was observed for *S. aureus* in the wounds treated by the gels incorporated with LP-G18-A11 or ciprofloxacin (Figure 3C). On the other hand, in the wounds contaminated with *P. aeruginosa*, reductions of 25% and 35% were observed for the gels containing LP-G18-A11 or ciprofloxacin, respectively (Figure 3D).

### 2.6. Evaluation of Formulation Stability

The gel containing 5% *L. plantarum* LP-G18-A11 was submitted to the centrifugation test. The gel remained unchanged during all three centrifugation cycles, without presenting phase separation, color change or the same change in color and odor. Therefore, it was suitable to perform the other stability tests.

#### 2.6.1. Preliminary Stability Test

In the preliminary test, the samples were evaluated for fourteen days, undergoing cycles of heating (on study at 45 ± 2 °C) and cooling (freezer at −7 °C). The gel containing *L. plantarum* LP-G18-A11 did not show significant changes in its physical-chemical and organoleptic characteristics, although variations were observed in the pH of the formulation on different days (Table 3).

#### 2.6.2. Accelerated Stability Test

In the accelerated stability test, the gel containing *L*. *plantarum* LP-G18-A11 was subjected to different storage temperatures, namely: room temperature (25 ± 2 °C), high temperature (37 °C) and low temperature (2 °C), for a period of 30 days. The organoleptic and physical-chemical characteristics are shown in Table 4. It was seen that, after 15 days, there was a slight change in the appearance of the gel when stored at 25 °C and at 37 °C, a fact that was not observed when it was subjected to the preliminary stability tests. A slight decrease in the pH was also seen over the days in all storage forms (Table 4).

## 3. Discussion

Skin infections are among the most common infectious diseases, ranked as the fourth cause of human disease [6,20,21]. The growing resistance presented by skin pathogens is considered to be the main problem in the treatment of skin infections in hospitalized patients. This results in a significant increase in morbidity and mortality as the usual antibiotics are not effective [22,23]. This scenario denotes the urgent need for new alternatives to treat or prevent this condition. In this context, this study aimed to develop a gel with lactobacilli with antimicrobial potential for dermatological use.

Firstly, we reported that all of the analyzed strains (*L. rhamnosus* ATCC 10863, *L. fermentum* ATCC 23271, *L. plantarum* ATCC 8014, and *L. plantarum* LP-G18-A11) inhibited the growth of *S. aureus*, while only *L. plantarum* LP-G18-A11 had a good result against *P. aeruginosa*. The antimicrobial effects of the lactobacilli strains are related to the release of bacteriocins and other bioactive molecules whose properties inhibit the growth of pathogenic microorganisms and/or interfere with the quorum sensing systems [24,25,26]. These bacteriocins and bioactive molecules may vary from one strain to another, which would explain the opposite results [27]. In fact, a wide range of studies have verified the antibacterial properties of lactobacilli strains against *P. aeruginosa* and *S. aureus*. For instance, two different studies showed that *L. plantarum* 34-5 showed antagonistic activity on clinical isolates of *P. aeruginosa* [28,29]. Other studies reported the antimicrobial and antivirulence properties of *L. plantarum* F-10 [30] and *L. plantarum* USM8613 against *S. aureus* [31]. Therefore, the use of a formulation holding probiotics has been proposed as an alternative to the use of antibiotics [32,33].

The four lactobacilli strains were incorporated into natrosol gels, a hydroxyethyl cellulose-based formulation with a suitable consistency and pH for application on the skin that has been used for the development of healing products [34,35]. In addition, the osmotic properties of a hydroxyethyl cellulose-based formulation allow it to absorb the liquids released from the wounded tissues [36,37]. The antimicrobial effects of the gels containing the lactobacilli were verified against *S. aureus* and *P. aeruginosa*. The best inhibitory activity was found for the *L. plantarum* LP-G18-A11-incorporated gels at 5% and 3%.

Subsequently, the viability of *L. plantarum* LP-G18-A11 in the gels was analyzed for three months. This test is of paramount importance, as one of the biggest challenges in using lactobacilli as the active ingredient is its viability in cosmetic formulations. This evaluation is critical to determine the shelf life of the formulation [38]. One strategy to improve the viability is the use of its lyophilized form, and even then, depending on the storage conditions, it may become unfeasible in a few days [39]. This approach was employed in our study and enabled the viability to remain for up to 14 and 90 days at 25 °C and 4 °C, respectively. In addition, the time-to-death curve test was performed with the *L. plantarum* LP-G18-A11-incorporated gel (5%), which totally inhibited *S. aureus* growth after 4 h.

It was observed that the probiotic associated in the formulation remained viable for a longer time when stored in the refrigerator, which is in agreement with the study that evaluated the viability of lactobacilli strains when stored at a low temperature [40]. Similar results were also observed in a vaginal gel formulation containing *Lactobacillus crispatus* ATCC 33197 for the prevention of gonorrhea, which confirmed its longer viability when stored at 4 °C, although its viability dropped dramatically after two weeks [41].

The antimicrobial action of the *L. plantarum* LP-G18-A11-incorporated gel (5%) was further analyzed in an ex vivo of a wound infection using porcine skin. This model is low-cost and represents an interesting alternative as porcine skin offers histological similarity to human skin [42,43], and it mimics how microorganisms can grow and develop in vivo [44,45]. In addition, porcine skin has been already employed to evaluate the antimicrobial activity of *L. plantarum* USM8613 against *S. aureus* [31].

Stability evaluations are essential for the development of cosmetic formulations, as the active principles that are incorporated into cosmetic vehicles can change their characteristics, causing instability and altering the quality requirements [13,46]. The gel that served as the cosmetic vehicle for this research is recommended by the National Health Surveillance Agency (ANVISA) of Brazil. This vehicle maintained the antimicrobial action of *L. plantarum* LP-G18-A11 and allowed its viability. The incorporation of this strain did not significantly change the gel characteristics. It is noteworthy that maintaining the stability of the product is essential to ensure its safety and efficacy, in addition to guaranteeing the content of the active ingredient in the formulation and estimating its useful lifespan [47].

The decrease in the pH of the formulations stored in different environments may be related to the production of some acid metabolite or the decomposition of some raw material during the heating process [48]. The changes that can occur in products, such as changes in the homogeneity and organoleptic characteristics (color and odor) may be indicative of possible physical-chemical changes that may be occurring in the product [46].

## 4. Materials and Methods

### 4.1. Obtaining and Activating the Microorganism

The strains *L. rhamnosus* ATCC 10863, *L. fermentum* ATCC 23271, *L. plantarum* ATCC 8014, *S. aureus* ATCC 27853, *K. pneumoniae* ATCC 700603, *E. faecalis* ATCC 29212 and *P. aeruginosa* ATCC 27856 were obtained from the Microbial culture collection of CEUMA University, where they are kept at −80 °C. The strain *L. plantarum* LP-G18-A11 was commercially purchased. The lactobacilli strains were activated in a MRS broth (De Man, Rogosa and Sharpe), while the other strains were activated in a BHI medium (Brain Heart Infusion Broth).

### 4.2. Antagonism Assay-Spot Overlay Assay

The antagonistic activities of the lactobacilli strains against the selected isolates (*S. aureus* ATCC 27853, *K. pneumoniae* ATCC 700603, *E. faecalis* ATCC 29212 and *P. aeruginosa* ATCC 27856) were performed using the spot overlay technique. Five microliters of bacterial suspensions (1 × 10^8^ CFU/mL) of each probiotic were spotted on the surface of the MRS agar plate, followed by incubation at 37 °C and 5% CO_2_. After 48 h, the lactobacilli colonies were overlayed with 15 mL of BHI agar (0.8% agar) containing the respective pathogenic strain at a 1 × 10^8^ CFU/mL. After 24 h incubation, the inhibition zones were measured and interpretated, determined by the formula suggested by Halder and Mandal [49]:IZ = (dnib − dspot)/2(1)
where “dinib” represents the diameter of the no-growth zone surrounding the “spot” and “dspot” denotes the diameter of the tested probiotic growth zone. For the scores of the growth inhibition results, the following were considered: without inhibition capacity when R < 2 mm; low inhibition capacity with “R” values of 2–5 mm, and high inhibition capacity with “R” values > 6 mm [49].

### 4.3. Probiotic-Based Gel Formulation

For the preparation of the gel, hydroxyethyl cellulose was used as a thickening agent, responsible for the viscosity of the formulation and which has non-ionic characteristics. Glycerin was used as the wetting agent, to prevent the product from losing water to the external environment. The handling was performed according to the Brazilian Pharmacopoeia [47], with adaptations.

All components of the formulation were weighed on a precision analytical balance in a container that was sterilized at 121 °C for 15 min. Sterile distilled water was heated to 70 °C for the subsequent melting of the humectant, and the thickening agent was gradually homogenized in water using a vortex until complete dissolution. After cooling, the lyophilized probiotics were homogenized using a vortex at concentrations of 3% (CFU/mL) and 5% (CFU/mL) in individualized formulations. The entire procedure was performed in a laminar flow cabin. The formulations were submitted to stability tests according to the tests proposed by the ANVISA cosmetic stability guide [47].

### 4.4. Antimicrobial Assays with Probiotic-Based Gel

#### 4.4.1. Agar Diffusion Test

Bacterial suspensions (100 μL; 1 × 10^8^ CFU/mL) of *S. aureus* ATCC 25923 and *P. aeruginosa* ATCC 27856 were added to 10 mL of Mueller-Hinton medium, stabilized at 45 °C. The culture medium was poured into 90 mm petri dishes (with 20 mL of medium per plate) and left until it solidified [50]. Next, 4 wells, 9 mm in diameter, were made with sterilized straws and filled with hydroxyethyl cellulose-based gels containing 5% or 3% of the respective lactobacilli strain. The plates were subsequently incubated at 37 °C for 24 h. The tests were performed in triplicates. After 24 h, the halos formed around the wells were analyzed.

#### 4.4.2. Time-Kill Curve

To determine the antimicrobial activity of the *L. plantarum*-incorporated gel under the influence of time, the methodology described in M26-A of the Clinical and Laboratory Standards Institute [51] was followed, with adaptation. Initially, an inoculum of *S. aureus* in MRS-MH medium (1:2) was adjusted to 1.5 × 10^8^ CFU/mL. After this adjustment, the inoculum was diluted at a ratio of 1:20 in MRS-MH broth, reducing the concentration to 5 *×* 10^6^ CFU/mL. Then, 3 mL of this was added to each flask containing 27 mL of MRS-MH broth medium. One of the flasks was supplemented with the *L. plantarum*-based gel, the other served as a positive control, containing only *S. aureus*. The flasks were incubated at 37 °C for 30 h, with the aliquots removed, and serially diluted and plated at times of 0, 2, 4, 6, 8, 24 and 48 h after the addition of the bacterial inoculum. The viability of *L. plantarum* was confirmed by plating (without counting) at all analysis times. The results were expressed in CFU/mL by counting the colonies after 20–24 h of plate incubation [14].

### 4.5. Viability and Storage Time of the Gel

The quantification of the viability of the lactobacillus strains was performed by plating the bacteria after serial dilutions in saline solution and plating in MRS agar medium, incubated for 48 h at 37 °C, and the storage time for 3 months was verified, with the plating every 7 days in the first month and monthly after the first month [52].

### 4.6. Evaluation of Formulation Stability

The preliminary evaluation consisted of two tests: centrifugation and thermal stress. The macroscopic aspects and viscosity were evaluated according to the Cosmetic Products Stability Guide, where: IM: moderate intensity; -M: modified; -LM: slightly modified; -B: normal, no change in appearance [47].

#### 4.6.1. Appearance and Homogeneity of the Formulation

Periodically, evaluations were conducted to evaluate changes in the color, odor, viscosity and homogeneity; the reference parameter was the product itself right after handling, in which all these initial parameters were recorded to serve as a guide. The physical characteristics of the sample were analyzed, placing the product in a 25 mL test tube and contrasting it against a dark background, as the gel color is light, according to the Brazilian Pharmacopoeia [47].

#### 4.6.2. Spin Test

In a 15 mL tube, 5 g of the sample was weighed and submitted to three cycles at 3000 rpm each for thirty minutes. The test was conducted in triplicate.

#### 4.6.3. Preliminary Stability

During the 14 days, the samples were submitted to a cycle of freezing and thawing (Table 5). The organoleptic, physical-chemical and chemical alterations were evaluated at time T0, T2, T4, T6, T8, T10, T12 and T14.

#### 4.6.4. Accelerated Stability

The samples were stored in a plastic container at room temperature, indirect light (25 ± 2 °C), refrigerator (5 ± 2 °C) and oven (37 ± 2 °C), and they were evaluated for three months, verifying the organoleptic, physical-chemical and chemical characteristics, with the analysis being read at time T0, T7, T15, T30 (28).

### 4.7. Assessment of Physicochemical Stability

The samples were stored at room temperature, indirect light (25 ± 2 °C), refrigerator (5 ± 2 °C) and oven (37 ± 2 °C) and were collected at time 0, 07, 15 and 30 days for the physical-chemical evaluation of the appearance, homogeneity, pH, viscosity, density and content of the active substance [47].

#### 4.7.1. Macroscopic Evaluation

During the storage, changes in the organoleptic characteristics (color, odor and appearance) and in the formulation state (phase separation, precipitation and turbidity) were analyzed.

#### 4.7.2. Determination of the pH Value

The pH was determined according to the aqueous dispersion of 10% (1:10) of the sample in distilled water and the measurement was made in a table pH meter, duly calibrated with standardized stock solutions. The aqueous dispersion containing the gel was placed in a 250 mL backer at room temperature (25.0 ± 2 °C) and the electrode was introduced into the solution.

#### 4.7.3. Density

Density was performed using an already calibrated 50 mL pycnometer. The density was determined by weighing the empty pycnometer, recording its value and subsequently determining the value of the pycnometer containing the test formulation. It is determined using the formula:Density (d) = (m sample − m empty)/(m water − m empty) (2)
where: m sample = mass of the pycnometer filled with the sample, m water = mass of pycnometer filled with distilled water, m empty = mass of empty and dry pycnometer.

### 4.8. Ex Vivo Model of Wound Infection

#### 4.8.1. Obtaining, Processing and Disinfecting Porcine Skin

The porcine skin was collected in a regulated slaughterhouse in the city of São Luís (Maranhão, São Luís). The skin sample was frozen until the experiment was conducted. The thawed skin was cut into 2 × 2 cm fragments using a scalpel and the disinfection process consisted of placing the skin fragments, with the aid of sterile tweezers, for 30 min in a sterile bottle containing 50 mL of 70% ethyl alcohol. Then, they were transferred to another sterile bottle containing 0.615% sodium hypochlorite for 30 min; finally, the skin fragments were placed in a sterile flask with 50 mL of distilled water for 30 min, then placed in sterile petri dishes for drying [43]. To form the wound, a punch (n° 8) was used to delineate the dimensions [45]. The punch was introduced into the dermis; then, with the aid of a scalpel, the demarcated tissue was removed.

#### 4.8.2. Wound Infection and Treatment

The skin fragments were individually placed in a Petri dish holding bacteriological agar at 7 g/L for stabilization and hydration. Bacterial inoculums were prepared according to McFarland’s 0.5 scale in falcon tubes containing BHI broth with 1% glucose. Later, 25 μL of bacterial inoculum was added to each wound. Then, the plates were placed at 37 °C. After 24 and 72 h, each wound was treated with the gel containing the probiotic. The skin fragments were used as the growth control, which were filled with gel only. All of the experiments were performed in triplicate [42,45].

#### 4.8.3. Quantification of Colony Forming Units (CFU)

For the CFU quantification, 50 µL of PBS was added to the wound bed and then 50 µL was aspirated. Subsequently, a cotton swab was introduced perpendicularly into the wound bed and rotated 360°, twice, and transferred to tubes holding 1 mL of saline + Tween 20 (5 µL/mL) for the detachment of the bacterial cells. (35) The tubes were vortexed for 1 min and 100 µL of each suspension was submitted to a serial dilution in sterile saline solution of 1:10; 1:100; 1:1000; 1:10,000 and 1:100,000. Subsequently, 10 µL of the 1:10,000 and 1:100,000 dilutions were seeded in duplicate on the selective medium (mannitol and Macconkey agar) and incubated at 37 °C for 24 h for CFU counting [42].

### 4.9. Statistical Analysis

The results were expressed as the mean ± standard error of the mean. The inhibition percentages were calculated with the mean of the inhibitions obtained for each individual experiment. The statistical evaluation of the results was performed using analysis of variance (ANOVA), followed by the Boferroni or Kruskal—Wallis test, for the parametric and non-parametric data, respectively, and a significance level of 0.05 was adopted.

## 5. Conclusions

The present study showed the effectiveness of using strains with probiotic potential in reducing the growth of *P. aeruginosa* and *S. aureus*, which are frequently found in wounds. *L. plantarum* LP-G18-A11, the strain with the highest antimicrobial potential, presented high viability in the formulated natrosol gel during storage and maintained its inhibitory effects. The incorporation of *L. plantarum* LP-G18-A11 also did not change the organoleptic and physical-chemical characteristics of the gel. Finally, it was proven that the LP-G18-A11-incorporated gel was able to reduce the bacterial load in an ex vivo wound model using porcine skin. Given the antimicrobial potential of the gel containing *L. plantarum* LP-G18-A11, further studies should improve the viability of the *L. plantarum* LP-G18-A11 in the gel, such as by optimizing the freeze-drying process or by adding some nutrients to the formulation. The optimized formulation should also be used in in vivo models of infected wounds to provide more insights into its antimicrobial action.

## Figures and Tables

**Figure 1 pharmaceuticals-16-00468-f001:**
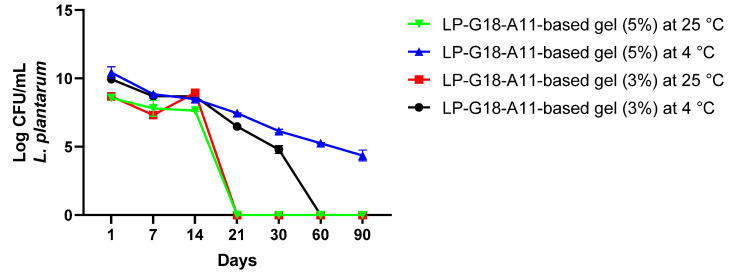
Viability of *Lactiplantibacillus plantarum* LP-G18-A11 at 3% and 5% in natrosol gels stored for 90 days at 4 °C and 25 °C.

**Figure 2 pharmaceuticals-16-00468-f002:**
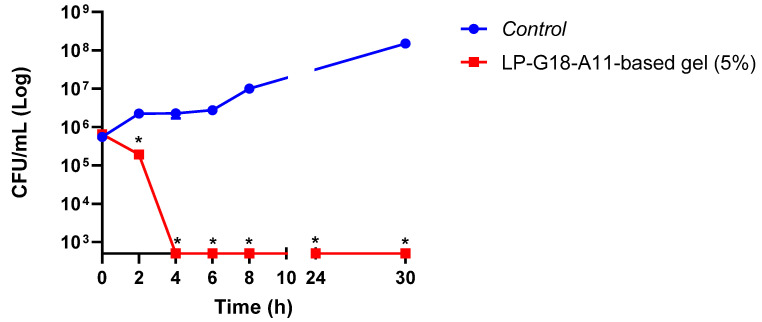
Time-death curve of *L. plantarum* LP-G18-A11-incorporated gel (5%) against *S. aureus*. * Significant difference in relation to the growth control (*p* < 0.05).

**Figure 3 pharmaceuticals-16-00468-f003:**
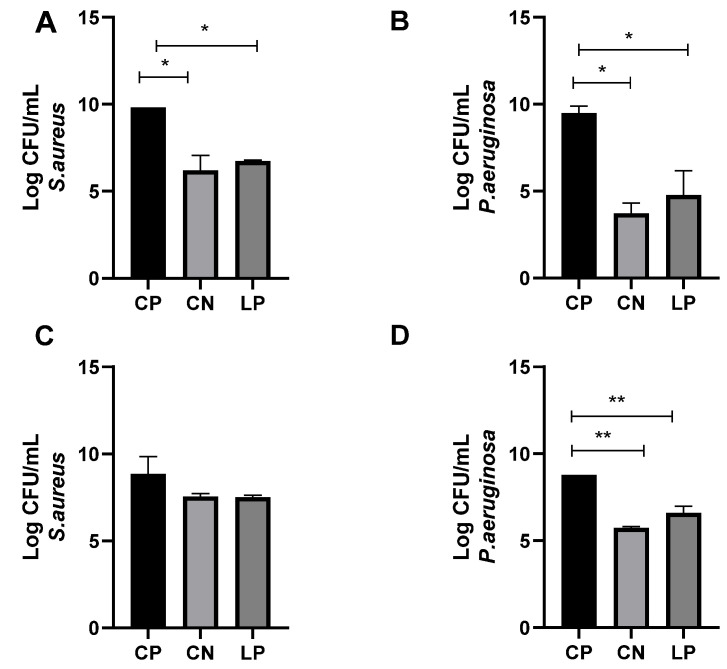
Evaluation of antimicrobial effects of *L. plantarum* LP-G18-A11-incorporated gel (5%) in an ex vivo model of skin wound. * *p* < 0.05; ** *p* < 0.01. CP: (**A**,**B**): Antimicrobial effects after 24 h. (**C**,**D**): Antimicrobial effects after 72 h. Positive control (*Staphylococcus aureus* or *Pseudomonas aeruginosa)*; CN: Ciprofloxacin; LP: *Lactiplantibacillus plantarum* LP-G18-A11.

**Table 1 pharmaceuticals-16-00468-t001:** Inhibitory activity of lactobacilli strains against pathogenic bacteria.

	Inhibition Zone (mm ± SD)			
Pathogens	*L. plantarum* LP-G18-A11	*L. fermentum* ATCC 23271	*L. rhamnosus* ATCC 10863	*L. plantarum* ATCC 8014
SA	12.5 ± 0.2 ^a^	8.0 ± 0.3 ^b^	8.0 ± 0.2 ^b^	8.0 ± 0.1 ^b^
KP	0 ^a^	0 ^a^	0 ^a^	0 ^a^
PA	8.5 ± 0.1 ^a^	0 ^b^	3.5 ± 0.3 ^c^	3 ± 0.2 ^c^
EF	0 ^a^	0 ^a^	0 ^a^	0 ^a^

SA: *Staphylococcus aureus*; KP: *Klebsiella pneumoniae*; PA: *Pseudomonas aeruginosa*; EF: *Enterococcus faecalis*. In each line, different superscript letters (^a, b, c^) show significant differences (*p* < 0.05).

**Table 2 pharmaceuticals-16-00468-t002:** Antimicrobial activity of natrosol gel with lactobacilli strains.

		Inhibition Zone (mm ± SD)			
Pathogens	*L. plantarum* LP-G18-A11 (5%)	*L. plantarum* LP-G18-A11 (3%)	*L. fermentum* ATCC 23271 (5%)	*L. rhamnosus* ATCC 10863 (5%)	*L. plantarum* ATCC 8014 (5%)
SA	10 ± 0.4 ^a^	8.0 ± 0.1 ^b^	0 ^c^	0 ^c^	0 ^c^
PA	17 ± 0.1 ^a^	5.0 ± 0.3 ^b^	0 ^c^	0 ^c^	0 ^c^

PA: *Pseudomonas aeruginosa*; SA: *Staphylococcus aureus*. In each line, different superscript letters (^a,b,c^) show significant differences (*p* < 0.05).

**Table 3 pharmaceuticals-16-00468-t003:** Organoleptic and physical-chemical characteristics of gel containing *L. plantarum* LP-G18-A11 in preliminary stability test.

Time (Days)	Organoleptic and Physical-Chemical Characteristics					
	Color	Odor	Appearance	Texture	Density	pH
T0	I	I	I	great texture, easy spreadability	0.902	3.8
T2	I	I	I	great texture, easy spreadability	0.900	3.6
T4	I	I	I	great texture, easy spreadability	0.890	3.6
T6	I	I	I	great texture, easy spreadability	0.900	3.3
T8	I	I	I	great texture, easy spreadability	0.886	3.3
T10	I	I	I	great texture, easy spreadability	0.880	3.3
T12	I	I	I	great texture, easy spreadability	0.890	3.1
T14	I	I	I	great texture, easy spreadability	0.889	3.1

Appearance: (I) normal, unchanged.

**Table 4 pharmaceuticals-16-00468-t004:** Evaluation of the organoleptic characteristics of gel containing *L. plantarum* LP-G18-A11 in accelerated stability test.

Temperature	Time	Organoleptic and Physical-Chemical Characteristics					
		Color	Odor	Appearance	Texture	pH	Density
25 °C	T0	I	I	I	great texture easy spreadability	3.8	0.902
	T7	I	I	I	great texture easy spreadability	3.4	0.900
	T15	I	I	II	great texture easy spreadability	3.0	0.887
	T30	I	I	II	great texture easy spreadability	2.8	0.800
4 °C	T0	I	I	I	great texture easy spreadability	3.8	0.902
	T7	I	I	I	great texture easy spreadability	3.5	0.901
	T15	I	I	I	great texture easy spreadability	3.0	0.905
	T30	I	I	I	great texture easy spreadability	3.0	0.900
37 °C	T0	I	I	I	great texture easy spreadability	3.8	0.902
	T7	I	I	I	great texture easy spreadability	3.4	0.905
	T15	I	I	II	great texture easy spreadability	3	0.907
	T30	I	I	II	great texture easy spreadability	2.9	0.799

Appearance: (I) normal. unchanged; Color and odor: (I) normal. without alterations; (II) slightly modified.

**Table 5 pharmaceuticals-16-00468-t005:** Temperature reference values for thermal stress tests.

Temperature Values
High-temperature values: 45 ± 2 °C
Low-temperature values: 5 ± 2 °C
Heating cycle values: 24 h at 45 ± 2 °C, and from 24 h: 5 ± 2 °C for 14 days

## Data Availability

The data are available upon request.

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
