# Peer review of "Development and Characterization of Hydroxyethyl Cellulose-Based Gels Containing Lactobacilli Strains: Evaluation of Antimicrobial Effects in In Vitro and Ex Vivo Models"

_pharmaceuticals, 2023, doi:10.3390/ph16030468_

Round 1

Reviewer 1 Report

The research work: "Development and Characterization of Topical Formulation Containing Probiotics: Evaluation of Antimicrobial Effects in In vitro and Ex Vivo Models" is novel and promising for future use. 

1. Material method should be first, then result and discussion. Even results and discussion can be given and discussed together, for better understating. 

2. What is the rationale behind topical delivery? Is Microorganism stable in a gel environment for a long time? 

3. Cite all the processes with recent latest work, if it is not innovative. 

4. Why is only one percent of gelling material used? should try higher as well as lower concentrations too, to study the effect of release.

Author Response

Dear reviewer,

Thank you so much for your valuable comments that improved the quality of our manuscript. Herein, we provide a point-to-point response for each of your comments. The changes performed are highlighted in yellow in the updated manuscript.

Response to Reviewer 1 Comments

The research work: "Development and Characterization of Topical Formulation Containing Probiotics: Evaluation of Antimicrobial Effects in In vitro and Ex Vivo Models" is novel and promising for future use.

  1. Material method should be first, then result and discussion. Even results and discussion can be given and discussed together, for better understating.

Response 1: Thank you very much for your considerations. However, we followed the template provided by the journal where the results and discussion are two separated sections and presented prior to Material and Methods.

  1. What is the rationale behind topical delivery? Is Microorganism stable in a gel environment for a long time?

Response 2: Thank you very much for your considerations.

The topical or local application of lactobacilli with probiotic potential has been exploited as an alternative for skin healing due the immunomodulatory and antimicrobial effects of these bacteria. For this, vehicles have been evaluated to promote the microorganism viability and allow the topical application. The viability of the bacterium is strain dependent. In the case of this study, natrosol gel was chosen. The polymer (hydroxyethylcellulose) is a cellulose derivative considered as thickening agent, responsible for the viscosity of the formulation and which has non-ionic characteristics.

  1. Cite all the processes with recent latest work, if it is not innovative.

Response 3: Thank you very much for your considerations.

A reference was added in the method of item 4.5 referring to the viability and storage time of the lactobacillus in the gel (reference number 45). In addition to this, a reference to the time of death curve method of the gel containing probiotic was added in item 4.4.2 (Reference number 44).

  1. Why is only one percent of gelling material used? should try higher as well as lower concentrations too, to study the effect of release.

Response 4: Thank you very much for your considerations.

The concentration of 1.5% was used because it is a concentration that gives the gel an incorporated average, which facilitates the application of the product on the skin and good spreadability and would keep it with a good accelerated application, without the possibility of “dripping” easily.

Reviewer 2 Report

This is a topic that is certainly relevant and interesting from the scientific point of view.

However, there are some explicit remarks that should be addressed prior to publishing the manuscript:

1. Bacteria strains should be written italic, also in vivo, in vitro, etc.

2. Introduction: In my opinion, this section should be a little bit more conciese. In particular, I suggest to rewrite the first two paragraphs (lines 35-46) which are very general. The focus towards real medical applicability, future perspectives (which are briefly mentioned in abstract due to antibiotic resistance) is given further and I believe, this is the key to increase the scientific value of this contribution.

As each study is based on the previous research, this should be indicated by a proper referencing, placed in the introductional section. This should also contain a wider overview of the other formulation methods and discuss their advantages and disadvantages. This part of the manuscript should be partially rewritten.

3. Section Materials and methods: Determination of Mic could give another valuable information of applicability of formulations.

4. A more extensive discussion and comparison to the available literature data (presented as a part of results with the corresponding deviation) is expected.

Minor revision is recommended.

Author Response

Dear reviewer,

Thank you so much for your valuable comments that improved the quality of our manuscript. Herein, we provide a point-to-point response for each of your comments. The changes performed are highlighted in yellow in the updated manuscript.

This is a topic that is certainly relevant and interesting from the scientific point of view.

However, there are some explicit remarks that should be addressed prior to publishing the manuscript:

  1. Bacteria strains should be written italic, also in vivo, in vitro, etc.

Response 1: Thank you very much for your considerations.These corrections have been made and are marked in yellow

  1. Introduction: In my opinion, this section should be a little bit more conciese. In particular, I suggest to rewrite the first two paragraphs (lines 35-46) which are very general. The focus towards real medical applicability, future perspectives (which are briefly mentioned in abstract due to antibiotic resistance) is given further and I believe, this is the key to increase the scientific value of this contribution.

As each study is based on the previous research, this should be indicated by a proper referencing, placed in the introductional section. This should also contain a wider overview of the other formulation methods and discuss their advantages and disadvantages. This part of the manuscript should be partially rewritten.

Response 2: Thank you very much for your considerations.

As suggested, we left the first two paragraphs more concise. We added some studies that demonstrate the broad use of lactobacilli in wound healing processes, scar reduction, as well as in the infections treatment of in vitro and in vivo. [lines 56-57]

The properties of Natrosol gel (Hydroxyethyl cellulose) and what favored the choice of this gel as a vehicle for the lactobacilo under study was also added properties [lines 68-74].

  1. Section Materials and methods: Determination of Mic could give another valuable information of applicability of formulations.

Response 3: Thank you very much for your considerations. We understand the reviewer point, however I believe that time-kill curve and ex-vivo assays provide the needed information about the antimicrobial action of the formulation.

  1. A more extensive discussion and comparison to the available literature data (presented as a part of results with the corresponding deviation) is expected.

Response 4: Thank you very much for your considerations.

Added a better discussion regarding the Natrosol gel (Hydroxyethyl cellulose) used in the present study [lines 216-220].

Reviewer 3 Report

Overall, this study is well-designed, appropiately described and discussed. 

Additionally, I have some comments for the authors:

How would it be possible to prolong the viability of the LP-G18-A11 in natrosol gel? 

According to your manuscript, currently it is 90 days only, in 4°C.

For wound healing gels osmotic properties of the gel is highly relevant. 

Please describe your thoughts about it and include it in the discussion.

Author Response

Dear reviewer,

Response to Reviewer 3 Comments

Overall, this study is well-designed, appropiately described and discussed.

Additionally, I have some comments for the authors:

1- How would it be possible to prolong the viability of the LP-G18-A11 in natrosol gel?

According to your manuscript, currently it is 90 days only, in 4°C.

Response 1: Thank you very much for your considerations.

As the number of LP-G18-A11 survived in the gel during storage, more studies are needed to preserve its viability over time. In this sense, hydrogel lyophilization, addition of prebiotics in the formulation or extemporaneous mixture of gel and LP-G18-A11 can be interesting strategies. As added in conclusion

2 -For wound healing gels osmotic properties of the gel is highly relevant. Please describe your thoughts about it and include it in the discussion.

Response 2:

The osmotic properties of polysaccharides-based gels, such as those with cellulose and derivatives,  allow them to absorb liquids released from the wounds. We have included this information in the discussion.